Accepted at the ICLR 2024 Workshop on AI4Differential Equations In Science

# MATHEMATICAL MODELING OF SPATIO-TEMPORAL DISEASE SPREADING USING PDES FOR MACHINE LEARNING

**Jost Arndt, Jackie Ma**
Department of Artificial Intelligence
Fraunhofer Heinrich Hertz Institute
Einsteinufer 37, 10587 Berlin, Germany
{jost.arndt, jackie.ma}@hhi.fraunhofer.de

## ABSTRACT

In this paper, we numerically solve a foundational PDE that describes the spatio-temporal spread of an infectious disease. We solve this PDE with various different epidemiological parameters on the domain of Germany and map the solutions onto geographical regions. This solution, in combination with geographical distances and adjacencies, serves as a dataset to train and validate various machine learning models on the task of epidemiological predictions. We evaluate the abilities of prominent models on this dataset to forecast the spatio-temporal spread of a simulated infectious disease, their robustness, and denoising capabilities. This evaluation undermines the importance of testing performance and robustness separately.

## 1 INTRODUCTION

A significant amount of Machine Learning research has recently been conducted on epidemiological data, especially on COVID-19 (Lalmuanawma et al., 2020; Rahimi et al., 2023; Heidari et al., 2022; Shorten et al., 2021). One principal application of this research is the forecasting of diseases, which is often seen in the context of time series prediction. However, the data often provides spatial information as well. Due to geographical conditions of administrative locations and their irregular structure (e.g., varying neighbors and distances), this data is often likewise irregular. Owing to the recent surge of interest in Graph Neural Networks (GNNs), their use appears to be a canonical choice to handle such irregular data, see Kapoor et al. (2020); Fritz et al. (2022)

However, the available infection data suffers from multiple flaws: As a result of inconsistent testing strategies (over time, across countries and diseases), the data quality and, consequently, its inherent value is often uncertain. Combined with the relatively short duration or limited case numbers of monitored diseases, and the challenges associated with data privacy, an accurate, comprehensive and systematic study and benchmarking of models is particularly challenging.

One of the historical foundations of mathematically modeling an infectious disease was developed nearly a hundred years ago (Kermack & McKendrick, 1927) and describes the behavior of two entities: Susceptible and Infected (SI) populations. Moreover, as part of a case study, the authors derived a foundational ODE that has led to much follow-up research. Furthermore, based on this model, mathematical modeling in epidemiology witnessed advancements by the introduction of more variables to the SI-equations, extending it by the recovered (SIR), vaccinated (SIVR), exposed (SEIR), or by introducing different age groups and creating larger systems of ODEs. Current research during the COVID-19 pandemic is often still based on the rather simple SIR equation, such as Dehning et al. (2020); Chang et al. (2021). However, due to the lack of spatial complexity in an ODE-based model, research has been conducted on the modeling of spatio-temporal spread of diseases using PDEs. In this regard, we will follow a PDE originally provided in Murray (2003). PDEs are capable of modeling phenomena in multiple dimensions, though they are generally more challenging to solve.

Concurrent with the epidemiological advances, the field of active research regarding GNN-based PDE-solvers advanced quickly (Brandstetter et al., 2022; Raissi et al., 2019). In this field some models and concepts that are seemingly suitable for epidemiological predictions have been developed.

In this work, we bring together ML-based research on PDEs, and ML-based epidemiology while addressing the lack of high-quality epidemiological data for a rigorous benchmarking and testing of machine learning models. Furthermore, we want to advance the use of PDE-based modeling of diseases. We will next briefly describe the PDE that is related to the described SI equation which we have numerically solved. In particular, the solutions of this PDE will form a synthetic dataset on which we will define three Machine Learning tasks that are relevant for an application in practice. We follow with a test of different commonly used models on those tasks and analyze the weaknesses and strengths of these models.

## 2 EPIDEMIOLOGICAL PDE - SYNTHETIC DATASET

To model the spatio-temporal spread of infectious diseases, we consider the following PDE from Murray (2003)

$$
\frac{\partial S}{\partial t} = -rIS + D\Delta S
$$
$$
\frac{\partial I}{\partial t} = rIS - \alpha I + D\Delta I,
$$

where $\Delta$ denotes the Laplace operator and the functions $S(x,t), I(x,t)$ describe the densities of the susceptible and infected population over space and time. Throughout this work we use a 2D spatial domain that is created based on Germany. Therefore $x$ is a 2D vector, not a scalar. The functions $\alpha(x,t), r(x,t)$, and $D(x,t)$ in the equation represent dynamics rising from pathogens and the population. More precisely, $r$ describes the transmission rate of the disease, $\alpha$ describes the duration of the disease, and $D$ describes the speed of the diffusion, i.e. movement of the population. Note that by setting $D = 0$ one receives the underlying SI compartment ODE, while setting $r = \alpha = 0$ leads to two heat equations.

The dynamics throughout our simulations are heuristically determined based on intuitively meaningful values, and not only vary from one simulation to another but are space- and time-dependent functions, respectively, mirroring different behaviors of the population during an infection wave. We simulate 25 different infection scenarios with the presented PDE by varying the dynamics on the domain, resulting in 9100 time-steps. We evaluate the solutions $S$ and $I$ on 400 locations, representing statistical regions in Germany (*NUTS3*-format), resulting in 400 trajectories. We use their geolocations and border information to create adjacencies and distances for adjacent regions.

We generate numerical solutions of these equations with the finite element library *deal.ii* (Arndt et al., 2023), using the Rothe method, an Euler scheme and a Newton method to obtain a comprehensive synthetic dataset for the later tasks. We initialize an uninfected population for all 25 simulations, i.e. $I(.,0) = 0, S(.,0) = 1$ with zero-flow Neumann boundary conditions. To start an infection, we disturbed the initial conditions slightly at a few fixed points in space and observed indeed an infection spreading across the full domain and over time.

In Fig. 1 we have visualized some time-steps from our simulated dataset, as well as the nodes and their adjacency, which are mapped back to their underlying geographical positions. Visibly an infection wave is traveling through a graph.

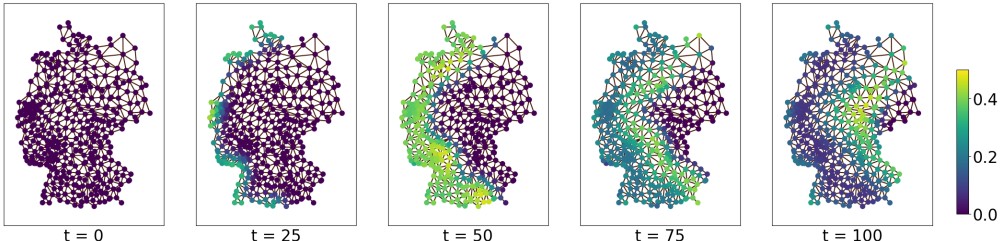

Figure 1: Plot of the 400 graph nodes and their adjacencies over time. The disease spreads from an infected population towards the susceptible population. Note, that the infection wave leaves areas of the graph being neither infected nor susceptible.

## 3 MACHINE LEARNING EXPERIMENTS

### 3.1 FORECASTING TASKS FOR DISEASE SPREADING

After solving the PDE, we exclusively use the value of the infections and drop any knowledge of susceptibility, to make the scenario closer to applications. Also, we removed any explicit information about the underlying PDE or its dynamics, and split it along the time-axis into a training and test dataset. To demonstrate the usage of our simulations the following three exemplary tasks, relevant for applications, are studied.

**Task 1: Baseline Experiment.** To evaluate different machine learning models, we generate predictions of the infection variable for the next 14 time-steps based on the preceding 14 steps. This is a common forecasting scenario.

**Task 2: Robustness.** As real-world data is known to be noisy, we decided to add some Gaussian noise to the input data during validation with the parameters $\mu = 0, \sigma^2 = 0.01$, and because infections can not take negative numbers or densities, we clipped the data at zero. The models are intentionally not retrained. This task suggests that the historical training data and recent testing data are different from one another, which we estimate to be a realistic scenario due to changes in testing strategies and changes in human behavior throughout time in real-world data. Consequently, we believe a study of the robustness of the different models holds significant value.

**Task 3: Denoising.** Finally, we test the capabilities of different models to intentionally denoise the input data. We retrain the models on data with the same Gaussian noise added as described above, then we evaluate how well different models can forecast 14 days of data, given a noisy input. Note, that only the input data contains noise, the targets stay unchanged. This would correspond to forecasting the actual number of infections given noisy measurements. We also estimate this to be a realistic and valuable scenario because historical real-world data likewise contains noise and the performance of models on disturbed data is of great interest.

### 3.2 MACHINE LEARNING MODELS FOR TIME SERIES

We continue with evaluating some prominent machine learning models that are frequently used for time-series forecasting. Our dataset especially incorporates spatial dimensions, therefore, we will additionally use mixtures of GNNs and time series models. Besides approaches coming from epidemiology, we will also incorporate an architecture from the community of GNN-based PDE solvers. Further details on these models can be found in as Supplementary Material B.2.

**Repetition.** As a simple baseline for a time-series forecasting challenge, we chose a naive repetition model. This model repeats the last input at every location as a forecast for the next one. There is no training involved.

**RNN.** Recurrent-Neural Networks based on GRUs Cho et al. (2014) have proven to be very successful for sequence prediction. This model takes a time series of context length 14 and generates a forecast over multiple days auto-regressively. During training, teacher forcing is applied.

**MP-PDE.** Given, that the underlying data is based on solutions of a PDE, we also test a model based on a Message-Passing PDE-solver (MP-PDE) from Brandstetter et al. (2022). Contrary to the original model, we will not pass any information of the underlying PDE to our model. The model consists of an encoder, a processor, and a decoder. The encoder creates node-wise embeddings of the context data. The processor consists of a Message Passing (MP) GNN, operating on a single graph with the embedded features. The decoder is a 1D convolution applied node-wise, and a special update rule, that propagates the last value through the next time-step.

**GraphEncoding.** We recreated a model from a recent contribution (Nguyen et al., 2023) that stems from the field of ML-based epidemiological forecasting. This model encodes the contextual time-steps separately in a shared GNN. The encoded graphs are then propagated through an LSTM (Hochreiter & Schmidhuber, 1997) network. To achieve a forecast for multiple days, this network is applied auto-regressively.

**RNN-GNN-Fusion.** Motivated by the PDE itself, we aimed to separate the time from the space dimensions and built an encoder RNN for the ODE, and a GNN to emulate the diffusion and combine both in a RNN decoder. The encoder is an RNN, analogously to the one in the abovementioned RNN model. Parallel to the encoder, an MP-GNN imitates the diffusion. The decoder consists of the same RNN as the encoder, but its forecasts and the GNNs output are combined in a convex combination. During training teacher-forcing is applied, and during validation, forecasts over multiple days are produced auto-regressively.

## 4 RESULTS

We tested all listed models in the three tasks explained in the previous section and plotted the RMSE for every time-step of the forecast individually. The plots of all experiments can be found in Fig. 2. The sum over the 14 time-steps can also be found as numbers in Table 1.

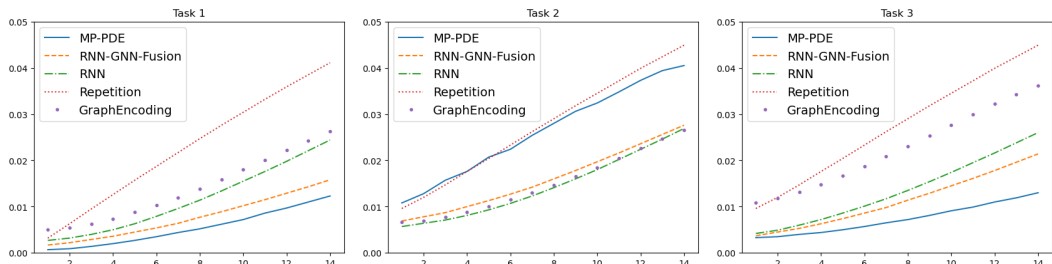

Figure 2: Comparison of models with depicted RMSEs. For each prediction step the RMSE is taken over all 400 regions and all samples of the test dataset

In Fig. 2 or Table 1 it is visible, that the spatial information can successfully be integrated and therefore two models incorporating GNNs exhibit the best performance in the first task. The MP-PDE performs the best while the RNN-GNN-Fusion shows good performance and abilities to incorporate diffusion into the recurrence. The GraphEncoding approach, however, seems to be only of limited benefit. The results from the second task suggest, that even though the MP-PDE outperforms other models on clean data, it is not very robust. The models that invoke an RNN right at their end all perform similarly, and better than the latter one. The results from the third task show, that MP-PDE can implicitly denoise data if it is already existent in the training dataset. The results are similar to the ones of the first task.

Table 1: RMSEs of the tested models over all 400 regions, and all samples in the test dataset summed up over 14 time-steps.

| Model | Task 1 | Task 2 | Task 3 |
|---|---|---|---|
| Naive repetition | 0.022793 | 0.027393 | 0.027393 |
| RNN | 0.011636 | **0.014414** | 0.013564 |
| GraphEncoding | 0.014007 | 0.014936 | 0.022543 |
| RNN-GNN-Fusion | 0.007714 | 0.015964 | 0.011379 |
| MP-PDE | **0.005414** | 0.026314 | **0.007329** |

## 5 CONCLUSIONS

Overall, we showed that the adoption of GNNs can advance the performance of epidemiological forecasting substantially. Nonetheless, we also showed that well-performing GNN architectures cannot be seen generally as the superior approach. In particular, our results suggest that robustness and performance are questions that deserve individual attention and are not guaranteed simultaneously. Also, we showed that the influence of AI research on PDEs can positively impact epidemiology regarding the architecture of models and suggest further research into pre-training and transfer onto real-world data.

With this work we want to emphasize the relevance of PDE based synthetic data to accelerate the advancement of AI models in applications.

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

## A    Acknowledgements

This work was supported by the Federal Ministry for Economic Affairs and Climate Action (BMWK) as grant DAKI-FWS (01MK21009A).

## B    Supplementary Material

### B.1    Epidemiological PDE

The domain and mesh are created with the help of *geopandas* (Jordahl et al., 2020) and *gmsh* (Geuzaine & Remacle, 2009). The underlying geographical data (*NUTS3*-shapefiles) of the domain, is taken from the Bundesamt für Kartographie und Geodäsie and is published under the dl-de/by-2-0 license.

### B.2    Models

The dataset is split only along the time-axis into $80\%$ training, and $20\%$ test data, on which we test the defined three tasks.

All models are created with the use of *PyTorch* (Paszke et al., 2019), GNNs furthermore with the help of *PyTorchGeometric* (Fey & Lenssen, 2019). The models are trained using the Adam optimizer (Kingma & Ba, 2015) and dropout (Srivastava et al., 2014).

**RNN.**

Recurrent-Neural Network architectures based on GRUs (Cho et al., 2014) have proven to be very successful for sequence prediction. This model takes a time series of context-length 14, and feature-size 1 (infected), and has 3 layers with each 64 hidden states. Right after the RNN, there is a MLP with two hidden layers with first dimension 64, then 1. During training, we use teacher-forcing and a dropout of 0.1 between each layer. To generate a forecast over multiple days during validation, the model is run autoregressively.

**MP-PDE.**

The model consists of three parts: an encoder, a processor, and a decoder. The encoder is an MLP with two hidden layers of size 128, transforming the context data of dimension 14 (context length) into a tensor of dimension 128. This is done on all 400 locations/vertices independently. The processor consists of 6 layers of a GNN, operating on a graph with 400 nodes with a feature size of 128, their adjacencies, their respective distances, and the unencoded data (skip connection). The GNN itself is a Message-Passing architecture, containing some MLPs. The decoder is a nodewise applied 1D convolution, to smooth the output along a time-dimension and a special update rule, that additively propagates the last value through the next time step. Further explanations and details can be found in the original paper (Brandstetter et al., 2022).

**GraphEncoding.**

This model interprets the 14 (context-size) time-steps in combination with adjacency and distances of 400 nodes as 14 distinct graphs and first processes these in a GNN separately. The GNN consists of two Graph Convolutional layers (Kipf & Welling, 2017) each with an output-size of 128, with a linear layer in-between, and ReLu and dropout layers between every layer. Both layer's outputs are concatenated, forming a skip connection around the second layer.

The output of the GNN therefore is a tensor of dimension $14, 400, 256$. On this tensor, two LSTM-layers (Hochreiter & Schmidhuber, 1997) are applied along the time axis, projecting the feature dimension from 256 to 128 to 128 combined with two more appending skip-connections around the second RNN layer and from the model's input. At the end, an MLP with two layers is applied, projecting the feature dimension from $256 + 14 = 270$ to 128 to 1. To achieve a forecast for multiple days, this network is applied autoregressively.

**RNN-GNN-Fusion.**

The encoder is built analogously to the pure RNN model above, with 3 layers with each 64 GRUs. Parallel, the last 7 input values for all 400 nodes, in combination with the nodes' adjacencies and

distances, propagate through a GNN. The GNN applies Message Passing and the message is constructed by a 3-layer MLP with a dropout of $0.2$, projecting from $14$ (infection value at the node concatenated with the difference to the neighboring value) to $64$ to $32$ to $1$ dimension. The resulting 1-dimensional output is weighted with the node's distance, which is also taken for normalization during the node update. The resulting scalar value should imitate the diffusion.

The decoder consists of the same RNN as the encoder, but its forecast for 1-timestep and the GNNs diffusion are combined in a convex combination.

During training teacher-forcing is applied, and during validation, forecasts over multiple days are produced autoregressively.

## B.3   RESULTS

The shown plots were created using *Matplotlib* (Hunter, 2007) and *pandas* (pandas development team, 2023).

