# OpenReview forum: "Mathematical Modeling of Spatio-Temporal Disease Spreading Using PDEs for Machine Learning"
_ICLR.cc/2024/Workshop/AI4DiffEqtnsInSci — AI4DiffEqtnsInSci @ ICLR 2024 Poster_

### Official Review · Reviewer_tJux · 2024-02-16
**Review of "Mathematical modeling of spatio-temporal disease spreading using PDEs for machine learning"**

**Rating:** 9
**Confidence:** 4

**Review:**

In this paper, the authors propose approximating a spatio-temporal PDE-based model of infection disease, based on tradition SI (Susceptible Infected) compartmental models typically used in epidemiology, with several data-driven / ML models.  Three experiments are designed, testing things like predictability, robustness and sensitivity to noise, and the proposed methods are evaluated using these experiments, on a test case which models the spread of disease within Germany.

The paper in question is very clear, concise and well-written.  I have not seen similar papers integrating PDE-based SI models with machine learning, and think there is some novelty here.  The experiments are well thought out and tackle a pseudo-realistic problem.  The information placed in the Appendix was appropriate for an appendix and nicely supplemented the main part of the paper.  Of the three ICLR papers I reviewed, this was by far the strongest.

Below are a few questions/corrections I had while reading the paper.  I realize that there is a page limit, and likely this is why some of the details I ask about were not provided.

- Section 2: I assume you are working with a 2D model of Germany, correct?  I suggest to state this explicitly.  I think in this case x should be a vector not a scalar.
- One question that may come up is why the PDE-based SI model cannot be solved on its own for projections without ML / data-driven models, if it is only a 2D model not discretized by a very fine grid.  It might be good to clarify this.
- How much work was required to train / optimize the NN-based models?
- Is there any insight into why the Graph Encoding approach does relatively well for task 2 but not for the other approaches?

---

### Official Review · Reviewer_sAP7 · 2024-03-01
**Synthetic epidemiological dataset generated could be useful for future research; Dataset generation and benchmarking experiments need more details.**

**Rating:** 6
**Confidence:** 4

**Review:**

Paper simulates an epidemiological dataset by solving a known set of PDEs for spatiotemporal evolution of diseases over a domain resembling Germany. The synthetic data is then used to benchmark various NN-based methods for forecasting.

1. More details on the dataset generation should be added to the paper:
	1. What are some examples of the 25 different infection scenarios?
	2. How were $\alpha(x, t), r(x, t), D(x, t)$ chosen?
	3. Was the dataset generated with different sets of initial points infected?
	4. How many trajectories does the dataset contain?
2. Benchmarking experiments need more clarifications/rigor.
	1. Please clarify whether the results in Figure 2 and Table 1 show RMSE for a single trajectory in the dataset?  Figure 2 is missing axis labels. Table 1 is missing confidence intervals.
    2. It would be helpful to show how the results change across the 25 different infection scenarios in the dataset (by varying $\alpha, r$, etc.)
	3. Robustness experiment is limited to a single level of arbitrarily chosen noise. It is unclear if the results hold across different levels of noise.
    4. Another relevant robustness scenario for epidemic modeling is when data is sparse/missing from certain regions of the domain. It could be interesting to benchmark whether the GNN based methods are robust to this.

Overall, novelty of the methodology is limited but a large-scale spatiotemporal epidemiological dataset could be useful for benchmarking or pretraining models.

---

### Meta-Review · Area_Chair_v7kr · 2024-03-01

**Recommendation:** Accept (Poster)

**Metareview:**

Authors present benchmarks with various ML models on forecasting spatiotemporal disease spread simulated from PDE-based epidemiological models. Reviewers have suggested adding more dataset details, improving the rigor of experiments and providing more insights into the comparative results between methods. For the camera-ready version, the authors should expand details on the data generation process and include additional information on model training and performance.

---

### Decision · Program_Chairs · 2024-03-02

Accept (Poster)